# Photon Counting CT Angiography of the Head and Neck: Image Quality Assessment of Polyenergetic and Virtual Monoenergetic Reconstructions

**DOI:** 10.3390/diagnostics12061306

**Published:** 2022-05-24

**Authors:** Arwed Elias Michael, Jan Boriesosdick, Denise Schoenbeck, Ingo Lopez-Schmidt, Jan Robert Kroeger, Christoph Moenninghoff, Sebastian Horstmeier, Lenhard Pennig, Jan Borggrefe, Julius Henning Niehoff

**Affiliations:** 1Department of Radiology, Neuroradiology and Nuclear Medicine, Johannes Wesling University Hospital, Ruhr University Bochum, 32429 Minden, Germany; jan.boriesosdick@muehlenkreiskliniken.de (J.B.); denise.schoenbeck@muehlenkreiskliniken.de (D.S.); ingo.lopez-schmidt@muehlenkreiskliniken.de (I.L.-S.); janrobert.kroeger@muehlenkreiskliniken.de (J.R.K.); christoph.moenninghoff@muehlenkreiskliniken.de (C.M.); sebastian.horstmeier@muehlenkreiskliniken.de (S.H.); jan.borggrefe@muehlenkreiskliniken.de (J.B.); julius.niehoff@muehlenkreiskliniken.de (J.H.N.); 2Institute of Diagnostic and Interventional Radiology, Faculty of Medicine and University Hospital Cologne, University of Cologne, 50923 Cologne, Germany; lenhard.pennig@uk-koeln.de

**Keywords:** computed tomography, photon counting detector, CT angiography of the head and neck, virtual monoenergetic reconstructions

## Abstract

Background: The purpose of the present study was the evaluation of the image quality of polyenergetic and monoenergetic reconstructions (PERs and MERs) of CT angiographies (CTAs) of the head and neck acquired with the novel photon counting CT (PCCT) method in clinical routine. Methods: Thirty-seven patients were enrolled in this retrospective study. Quantitative image parameters of the extracranial, intracranial and cerebral arteries were evaluated for the PER and MER (40–120 keV). Additionally, two radiologists rated the perceived image quality. Results: The mean CTDI_vol_ used in the PCCT was 8.31 ± 1.19 mGy. The highest signal within the vessels was detected in the 40 keV MER, whereas the lowest noise was detected in the 115 keV MER. The most favorable contrast-to-noise-ratio (CNR) and signal-to-noise-ratio (SNR) were detected in the PER and low keV MER. In the qualitative image analysis, the PER was superior to the MER in all rated criteria. For MER, 60–65 keV was rated as best image quality. Conclusion: Overall, PCCT offers excellent image quality for CTAs of the head and neck. At the current state, the PER of the PCCT seems to be the most favorable reconstruction for diagnostic reporting.

## 1. Introduction

CT angiography (CTA) of the head and neck is part of the standard diagnostic workup for acute ischemic stroke [1], as well as intracranial hemorrhage [2], and thus a standard examination in neuroradiology. In recent years, various technical innovations have contributed to an improvement of the image quality of CTAs. The introduction of the dual energy CT (DECT) [3,4], for example, offered the option of computing virtual monoenergetic reconstructions (MERs) using spectral data in addition to reconstructing polyenergetic images (PERs) [5]. MERs are generated in order to depict an image as it would have been acquired with a monochromatic X-ray beam [6]. MERs can be calculated in a wide range of energy levels, which are expressed in kiloelectron volts (keV) [7]. MERs can be used according to their respective strengths, which are relevant in neurovascular imaging: MERs with low keV levels provide increased soft tissue and iodine contrast, allowing for enhanced visualization of brain lesions and vessels [3,5,8], while higher keV levels can be used to reduce beam hardening artifacts; e.g., in order to improve vessel assessment after neurovascular interventions [9,10].

In 2021, the first photon counting CT (PCCT; Naeotom Alpha, Siemens Healthineers, Erlangen, Germany) approved for clinical use became available [11,12]. The introduction of photon counting detectors (PCDs) is considered to be the latest revolution in clinical computed tomography [13]. In conventional CT detector systems—also called energy integrating detectors (EIDs) or scintillator detectors—the X-ray radiation reaching the detector is converted into a light signal by a scintillator, which is then measured by a photodiode. In contrast, photons hitting the PCD directly generate electrical signals in a cadmium telluride semiconductor. Here, the electrical signal corresponds exactly to the energy of the absorbed photon [14]. Based on this technological innovation, PCDs are expected to provide spectral data with higher resolution and lower noise [15]. Initial studies on CT angiographies conducted with prototype PCCTs suggest that image quality, and thus diagnostic accuracy, is superior to CT systems using EIDs [16].

The purpose of the study presented here was to assess the image quality of PERs and MERs of CT angiographies of the head and neck conducted with the first PCCT in clinical routine. The assessment was based on well-established qualitative and quantitative image criteria. The results of this study are discussed and compared with results from studies using sophisticated DECT.

## 2. Materials and Methods

### 2.1. Patient Population

The study was conducted according to the guidelines of the Declaration of Helsinki and approved by the institutional review board. Patient consent was waived due to the retrospective study design.

Patients who received PCCT angiography of the head and neck between January and February of 2022 were included in this study. All CT examinations were performed because of a clinical indication, usually with the clinical suspicion of an acute stroke. Examinations with pronounced artifacts were excluded (e.g., motion artifacts or extensive beam hardening artifacts). Furthermore, patients with stents in the arteries of the head or neck, as well as patients with bilateral occlusions of an extracranial or large intracranial vessel, were excluded from this study.

### 2.2. CT Protocols and Image Acquisition

All CT scans were conducted with a clinical, approved PCCT (Naeotom Alpha, software version Syngo CT VA40, Siemens Healthineers, Erlangen, Germany) using a predefined, clinical scan protocol. All CT examinations were performed with the following acquisition parameters: the tube voltage was 120 kVp with automatic tube current modulation (IQ level 145, average effective current time product of 86.26 ± 13.39 mAs). The pitch was set at 0.8 with a rotation time of 0.25 s. The total collimation was 57.6 mm with a single collimation of 0.4 mm. The matrix was 512 × 512 and the field of view (FOV) was adjusted for each patient to optimally image the vessels from the aortic arch to the vertex. Contrast agent (70 mL, ACCUPAQUE^®^ 300, GE Healthcare, Chicago, IL, USA) was injected into a peripheral vein of the arm with the help of a coupled automated injector system (MEDRAD^®^ Centargo, Bayer Healthcare, Leverkusen, Germany) with a flow of 3.8 mL/s followed by a NaCl chaser (40 mL). Image acquisition was started with a delay of 8 s after reaching a threshold of 100 HU in an ROI placed in the ascending aorta.

A manufacturer-specific spectral workstation (Syngo.Via, VB60 version, Siemens Healthineers, Erlangen, Germany) was used to analyze the datasets. The axial polyenergetic and monoenergetic reconstructions were reconstructed from the same data set, applying the manufacturer’s specific algorithm (vascular kernels, Bv44 for polyenergetic images, Qv44 for monoenergetic images). The slice thickness was 1 mm and the slice increment was 0.6 mm. The quantum iterative reconstruction level Q2 was used for all images.

### 2.3. Quantitative Image Analysis

For quantitative image analysis, virtual monoenergetic reconstructions (MERs) from 40 keV to 120 keV in 5 keV steps were reconstructed in addition to the polyenergetic reconstruction (PER). Analogously to the method used by Neuhaus et al. [17], nine different regions of interest (ROIs) were set to assess the quantitative image quality. Segments of the vessels were named following Bouthillier’s classification (Bouthillier, van Loveren, & Keller, 1996) [18].

Two ROIs were placed in the large extracranial vessels below the skull base: one in the internal carotid artery in the extracranial segment (segment C1), the other in the vertebral artery in the atlantic extradural segment (V3). The next two ROIs were placed in the major intracranial brain-supplying vessels: in the internal carotid artery in the cavernous segment (C4) and in the basilar artery. Further ROIs were placed in each of the proximal cerebral arteries: the middle cerebral artery (MCA) in the M1 segment, the anterior cerebral artery (ACA) in the A1 or proximal A2 segment, and the posterior cerebral artery (PCA) in the P1 or proximal P2 segment. Finally, two ROIs were placed in the lateral pterygoid muscle and in the air immediately adjacent to the patient’s neurocranium. 

The size of the ROIs within the vessels was chosen to be as large as possible, ensuring that only the contrasted lumen of the artery was measured. The ROIs in the lateral pterygoid muscle as well as the ROIs in the adjacent air were set with a constant size of 25 mm^2^. Both the size and position of the ROIs were kept exactly the same between reconstructions, adapting the approach of Neuhaus et al. [17].

Signal was defined as the average density of voxels of the ROIs in Hounsfield units (HU), and noise was defined as the standard deviation (SD) of all voxels of the ROI. The signal-to-noise ratio (SNR) of an ROI was calculated as the mean density divided by the SD of the ROI. In analogy to previous studies [17,19], the contrast-to-noise ratio (CNR) of a ROI was calculated as follows:$$\mathrm{CNR}=\frac{signal_{artery}-signal_{muscle}}{standard\;deviation_{air}}$$

For the quantitative analysis, data from the individual ROIs were first screened for the presence of different trends described in other studies; for example, for the basilar artery [17]. After determining that behavioral patterns of the arteries (with respect to the quantitative parameters) had been excluded, the data from the ROIs in the extracranial internal carotid artery and extracranial vertebral artery were combined as data from extracranial arteries. Similarly, the data from the intracranial internal carotid artery and basilar artery were combined as data from intracranial arteries and the data from the ACA, MCA, and PCA were put together as data from cerebral arteries. The individual image quality parameters were then examined for these groups of vessels.

### 2.4. Qualitative Image Analysis

The image quality of PERs and certain MERs (40 keV, 50 keV to 80 keV in 5 keV steps, 90 keV, 100 keV, 120 keV, for a selection see Figure 1) was evaluated by two experienced radiologists with 17 and 12 years of experience, respectively, using 5-point Likert scales. The internal carotid artery in the region of the skull base and intracranial (from distal segment C1 to C7) as well as the distal vertebral artery and the basilar artery were rated with regard to the image quality for diagnostic evaluation. In particular, the region of the internal carotid artery with typical calcifications was rated on a scale from 1 = “difficult, uncertain diagnosis” to 5 = “excellent, fully diagnostic”. Next, the image quality for diagnostic evaluation of the small vessels (A3 and distal, M3 and distal, P3 and distal) was evaluated on a scale from 1 = “poorly defined, uncertain diagnosis” to 5 = “excellent delineation, fully diagnostic”. Finally, contrast was rated on the scale from 1 = “poor contrast” to 5 = “excellent contrast”, and noise was rated on the scale from 1 = “excessive noise” to 5 = “none”. Readers were strongly encouraged to repeatedly adjust the window settings.

### 2.5. Statistical Analysis

Data processing and statistical analyses were performed with the statistical software R and RStudio (R Core Team (2021)). R: A language and environment for statistical computing. R Foundation for Statistical Computing, Vienna, Austria. RStudio Version 1.4.1106). The Shapiro–Wilk test was used to test for normal distribution. Differences in normally distributed variables were tested by means of the two-sided paired *t*-test. The two-sided Wilcoxon signed-rank test was used to test for differences in variables that were not normally distributed.

For the purpose of qualitative analysis, an interval scale-level was assumed for the Likert scales. The two-sided Wilcoxon signed-rank test was used. *p*-values ≤ 0.05 were considered as statistically significant. If not stated otherwise, all data are presented as means ± SD.

## 3. Results

### 3.1. Patient Population and Radiation Dose Parameters

After identification of all CT angiographies of the head and neck conducted with the PCCT in the above mentioned time period and exclusion of the examinations following the above described criteria, 37 patients were included in this study. Of these patients, 20 were women and 17 men. The mean age was 74.9 ± 12.6 years. The majority of CTAs (*n* = 36) were performed when an acute stroke was clinically suspected, and one CTA was performed to identify the source of bleeding in a case of atypical parenchymal hemorrhage. The mean CTDI_vol_ was 8.31 ± 1.19 mGy with a mean scan length of 415.80 ± 103.73 cm. The average dose length product was 361.00 ± 69.83 mGy*cm.

### 3.2. Objective Analysis

#### 3.2.1. Signal

In the PER, the signal (Figure 2) from the extracranial brain-supplying arteries (EAs) was 377.15 ± 78.26 HU. In low keV MER, the signal was much higher (937.55 ± 204.71 HU at 40 keV). Across the MER, the signal dropped monotonically with increasing keV, being lowest in the 120 keV MER (113.74 ± 21.74 HU). The differences between MERs and the PER were all statistically significant. The signal in 40 keV to 60 keV MER images was significantly higher compared to the PER. At the same time, the signal in 65 keV to 120 keV MER images was significantly lower compared to the PER (*p* = 0.004).

The signal of the intracranial arteries (IAs) and cerebral arteries behaved analogously to the signal of the EAs. In comparison to the EAs, however, the signal was slightly lower in the PER (342.76 ± 70.77 HU, *p* = 0.009), as well as in each MER. In the low keV MER (up to 65 keV), the signal was higher and in higher keV MERs (70 keV and above) the signal was lower compared to the PER (PER 301.19 ± 62.51 HU, *p* < 0.001).

In the cerebral arteries (CAs), the signal in the PER was 323.68 ± 75.67 HU (compared to the EAs *p* < 0.001), while the low keV MERs (up to 60 keV) had a higher signal. Higher MERs (65 keV and above) had a significantly lower signal compared to the PER (319.35 ± 74.03 HU at 65 keV MER, *p* = 0.021).

#### 3.2.2. Image Noise

The image noise (Figure 3) was closely related to the level of the signal. In the MER it was highly pronounced at low keV levels, decreasing monotonically with increasing keV levels. In the EA, the noise was 12.23 ± 3.58 HU in the PER, and in the MER it dropped from 30.50 ± 12.73 HU in the 40 keV MER to 11.43 ± 3.38 HU in the 120 keV MER. The noise was significantly lower in the PER compared to MER up to 65 keV MER (*p* = 0.008). The difference in noise between PER and MER at 70 to 120 keV was not significant (*p* > 0.064).

Overall, the noise in the IAs was slightly higher than in the EAs (e.g., 14.18 ± 4.34 HU in the PER). The noise in the low keV MERs (up to the 65 keV) was significantly higher compared to the PER, and at 70 and 75 keV there was no significant difference. In the 80 keV MER (and above) the noise was significantly lower in the MER and it reached its minimum in the 110 keV MER (12.43 ± 3.60 HU).

In the CAs, the image noise in the PER was 13.88 ± 6.22 HU. In low keV MER the level of noise in the CAs was higher than in the EAs, but not as high as in the IAs. In the 70 keV MER (and above), the noise was less pronounced and it reached its minimum in the 115 keV MER (9.23 ± 4.61 HU).

#### 3.2.3. Signal-to-Noise-Ratio

For the EAs, the SNR (Figure 4) in the PER was 33.03 ± 10.54. For the MER at 40 and 45 keV, the SNR was slightly higher, but not significantly increased compared to the PER. Above the 50 keV MER, the SNR was lower than in the PER. Above the 55 keV MER, the differences between PER and MER were statistically significant (*p* = 0.034 at MER 55 keV). Likewise, in the IAs, the difference between the SNR in the PER (26.73 ± 10.50) and the MER from 40 to 50 keV was not statistically significant. However, the SNR in the 55 keV MER was significantly lower compared to the PER.

For the CAs, the SNR in the PER was 28.60 ± 16.97 (Figure 5). The differences in SNR between the MER with 40 to 75 keV and the PER were not statistically significant, and the dispersion of the values was substantial. Above the 80 keV MER, the SNR was significantly lower than in the PER (mean SNR at 80 keV 28.01 ± 30.97, *p* = 0.013).

#### 3.2.4. Contrast-to-Noise-Ratio

The CNR (Figure 6) reached the highest level in the EAs and the lowest level in the CAs. In the EAs the CNR in the PER was 58.72 ± 19.90. In the 40 keV MER the CNR was similar (59.01 ± 22.68, *p* = 0.447) to the PER. In the 45 keV MER (and above) the CNR was significantly lower compared to the PER (*p* < 0.001 for each keV level). This finding also applies to the CNR of the IAs, with a CNR in the PER of 52.25 ± 17.81 (40 keV 53.67 ± 21.68, *p* = 0.775; 45 keV 48.35 ± 19.27, *p* = 0.003; 50 keV and above *p* < 0.001). Regarding the CAs, the CNR reached its maximum in the PER with 48.53 ± 17.29. The CNR was significantly lower in all MER above 45 keV.

The most important data for the polyenergetic and the monoenergetic reconstructions are summarized in Table 1.

### 3.3. Qualitative Analysis

The PER was rated superior compared to the MER in all assessed criteria (Table 2, Figure 7). In the assessment of the large intracranial vessels, the 50 keV MER was rated best among all MERs with a mean score of 4.54 ± 0.53, which was still significantly inferior to the PER (4.82 ± 0.42). The small intracranial arteries were rated best in the MER at 65 keV (4.39 ± 0.66), which was again significantly lower compared to the PER (4.91 ± 0.41). In terms of general contrast, there was no significant difference between the 40 keV MER (4.76 ± 0.43) and the PER (4.85 ± 0.36, *p* = 0.057). However, in the 45 keV MER this difference was statistically significant (*p* = 0.036). Noise was rated as more distinct in both the low keV MER and relatively high keV MER. The best rated MER (60 keV, 4.28 ± 0.75) was still significantly inferior to the PER (4.81 ± 0.43, *p* < 0.001). Table 2 summarizes the most important data from the qualitative analysis.

## 4. Discussion

The present study evaluated the performance of the first clinical photon counting CT (PCCT) in high-contrast angiographies of the head and neck by assessing the quantitative and qualitative image quality. For this purpose, we compared the objective image parameters as well as the subjective image criteria of PER and MERs at different levels on the PCCT scanner. The choice of quantitative parameters—signal, noise, SNR and CNR—as well as the criteria for the qualitative assessment of the image quality were adapted from previous studies to ensure caution in the first comparisons [17,18,19,20,21,22,23,24].

In contrast to previously published studies based on dual energy CTs (e.g., [25,26,27]), the key finding of the present study was that at the current stage, considering all parameters together in photon counting CTA of the head and neck, the PER performed better than MERs in terms of quantitative as well as qualitative image quality. With respect to each individual quantitative parameter, an equivalent or superior MER compared to the PER was found in each case; these particular MERs were, however, clearly inferior to the PER with respect to the remaining parameters. The PER, on the other hand, combined excellent contrast (the strength of low keV MER) with low noise (the strength of the high keV MER). Therefore, as shown by the qualitative analysis, the PER currently seems to be the most favorable PCCT reconstruction for clinical reporting and provides dedicated high resolution in the depiction of small vessels. Further advances in image post-processing of PCCT data, however, may lead to an improvement of monoenergetic images and thus may allow for a stronger reduction of contrast medium to be used in PCCT angiography due to the higher inherent contrast. Consequently, it appears that it would be advantageous for further developments of post-processing to focus on both reducing noise in low keV MER, for better examination with dedicated low application of contrast media, and improving the resolution and imaging post-processing in PER, for optimal depiction of vascular malformations, aneurysms and dural arteriovenous fistulas. As a limitation, it has to be noted that the second level of quantum iterative reconstruction was used in this study; higher levels could possibly improve the MER further, since noise seems to be a major constraint here. Similarly, a soft kernel could lower the noise in the MER more than in the PER.

Recent studies have already investigated the image quality of CT angiographies in photon counting [16,28,29], especially with regard to comparison with conventional CT systems with EIDs. Euler et al. [28] investigated the image quality of photon counting CT angiography of the aorta, also comparing PER with a selection of low keV MERs. In this study the MERs of photon counting CT were more clearly superior to PER in quantitative parameters; however, a different definition of image noise partially complicates the comparison (see the Appendix A for a detailed comparison and discussion). In the qualitative assessment by Euler et al. it was shown, comparably to our study, that the readers saw increased noise in the low keV MERs compared to the PER.

The data acquired in our study are not sufficient for a direct comparison of the image quality with previous sophisticated CT systems such as DECT [17,19]. However, a cautious comparison of the results can be attempted to give a first impression of the performance of PCCT regarding the image quality of CTA of the head and neck. In particular, the comparison with advanced detector-based dual-layer spectral CT systems suggests that the image quality of the PCCT could be similar. Neuhaus et al. [17] describe a maximum CNR of 49.16 ± 8.57 in dual-layer CT (DLCT) in the 40 keV MER averaged over all vessels (proximal to distal). This compares to a CNR of 52.50 ± 18.14 (PER) averaged over all arteries and up to 58.72 ± 19.90 (PER) in extracranial arteries in the present study. The larger standard deviation in the present study can be attributed to outliers, particularly in the cerebral arteries. When comparing both studies, it should be noted that Neuhaus et al. used more contrast medium in their DLCT study compared to the present study (80 mL, 350 mg/mL versus 70 mL, 300 mg/mL). Most importantly, the radiation dose used by the PCCT (CTDI_vol_ 8.31 ± 1.19 mGy) in the current protocols was already considerably lower than in the other study (CTDI_vol_ 16.3 mGy) [17]. This cautious comparison does not allow proof of superiority of PCCT but offers the prospect of demonstrating in further studies that equivalent image quality can be achieved with a lower volume of contrast agent and a lower radiation dose.

As shown in the present study, the signal from the vessels decreased with decreasing vessel diameter, which is consistent with previously published data [17,19]. This was most likely related to a reduced concentration of the contrast agent within the peripheral vessels. Early acquisition of the images offers the advantage that cerebral veins and sinuses are poorly contrasted and do not complicate the assessment of brain-supplying arteries, which is the main focus of the CT examination.

Regarding the noise in the different vascular sections, it should be considered that the large intracranial arteries are very close to the skull base. The ROIs for measurement were placed in the C4 segment of the internal carotid artery, i.e., in the cavernous portion adjacent to the midline-forming section of the sphenoidal bone, and in the basilar artery, i.e., closely to the clivus. The proximity to the bone structures with consecutive tendencies for increased artifacts can potentially explain the increased noise compared to the EAs [17]. The higher noise in the CAs compared to the EAs may be explained by increased partial volume effects [30] in the small cerebral arteries. During data acquisition, only the contrasted lumen was explicitly measured. However, with very small artery diameters, partial volume effects are more likely. With increasing keV levels in MER, iodine contrast decreases and partial volume effects become less relevant so that the noise decreases even more. The differing image quality of vessels depending on the proximity to bony structures is a possible starting point for future studies, especially since in the present study proximity to bony structures for the grouping of the measured sections of the arteries was chosen to aid comparability with other studies.

It is known that the sensitivity of PCDs to X-rays (Figure 8) from the low energy spectrum is higher than the sensitivity of scintillator detectors [12,15]. This advantage in sensitivity in the low energy spectrum provides higher soft tissue as well as iodine contrast, as both—soft tissue and iodine, with its k-edge at 33 keV [31]—absorb more low energy photons than high energy photons. This sensitivity of the PCD could be one reason for the good performance of the PER since it increases the signal of soft tissue and iodine contrast. As in DECT, in the MERs of the PCCT, the signal within the contrasted arteries was significantly higher in the 40 keV MER than in the PER. However, the noise also increased to the same extent so that the MER did not perform better compared to the PER in terms of SNR and CNR in most cases. It is expected that the post-processing of the MER will be improved in the near future and that the noise can be further minimized [8]. Considering the qualitative analysis of the present study, the MERs do not yet offer improvements for clinical reporting.

The current study has limitations due to the retrospective approach and a possible minor selection bias. The group of patients with clinical suspicion of acute stroke was very heterogeneous, and the diversity of the different subgroups could not be adequately represented in this study. Furthermore, an important objection is that, based on the presented results, it cannot be concluded beyond doubt which reconstruction is suitable for detecting a particular pathology. After considering additional information and the follow-up diagnostics of the patients included in the present study (such as angiography and MRI), it can be concluded that photon counting CTA neither missed nor falsely detected a finding or diagnosis that was generally thought to be made or ruled out using CTA. However, in combination with the qualitative and quantitative analyses of similar studies (e.g., [17,19,30]), this first study evaluating the performance of clinical PCCT angiography is representative in addressing the first clinical and scientific observations.

PCCT is considered to be a major development in CT technology. An increase in resolution by a factor of 5 to 8 in comparison to conventional CT imaging has recently been reported [32]. At the same time a significant reduction of the radiation dose and contrast media is expected. In particular, this applies to high contrast structures, such as in CT angiographies as well as lung and bone imaging, which, in turn, may allow more precise diagnostics. This has previously been demonstrated for CT coronary angiography [28,31,32,33]. The capabilities of the PCCT in enabling the diagnostic evaluation of small intracranial vessels (Figure 9) are becoming increasingly important, especially due to the growing intra-arterial treatment options. The treatment of acute strokes, the diagnostic work-up of dural arteriovenous fistulas or malformations [34,35], the discrimination between aneurysms and infundibular vessel bifurcations, and post-therapeutic imaging, among other applications, would significantly benefit from improved imaging [36,37].

We conclude that in PCCT angiography of the head and neck, PERs are the most favorable reconstructions for diagnostic reporting at the current state. However, upcoming developments in post-processing may improve MERs, so the superiority of the PER must be critically re-evaluated in upcoming studies. Cautious comparison with older data from the literature on dual layer CT suggests that in CTA of the head and neck similar image quality can be achieved in terms of quantitative parameters, most likely with a lower volume of contrast agent and a lower radiation dose. The results of the present study underline the great potential of this novel detector technology to improve the image quality of CT angiographies. This, in turn, may lead to the use of less contrast agent and a lower radiation dose, which should always be an important, superordinate goal of advances in CT technology.

## Figures and Tables

**Figure 1 diagnostics-12-01306-f001:**
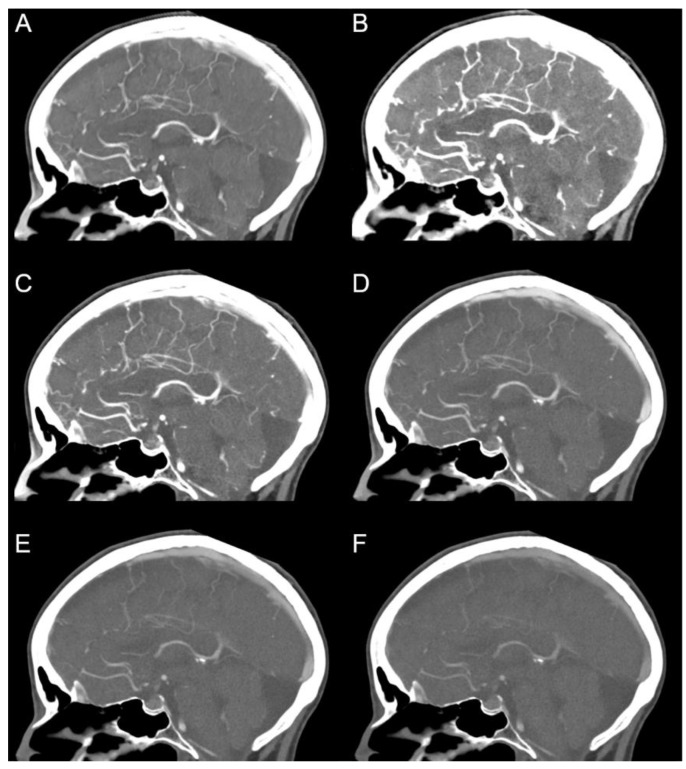
Polyenergetic and monoenergetic reconstructions. (**A**) polyenergetic reconstruction; monoenergetic reconstructions with (**B**) 40 keV, (**C**) 60 keV, (**D**) 80 keV, (**E**) 100 keV and (**F**) 120 keV. Window settings were identical in each image. Note the sharp visualization of the small intrasulcal vessels in the polyenergetic reconstruction (**A**).

**Figure 2 diagnostics-12-01306-f002:**
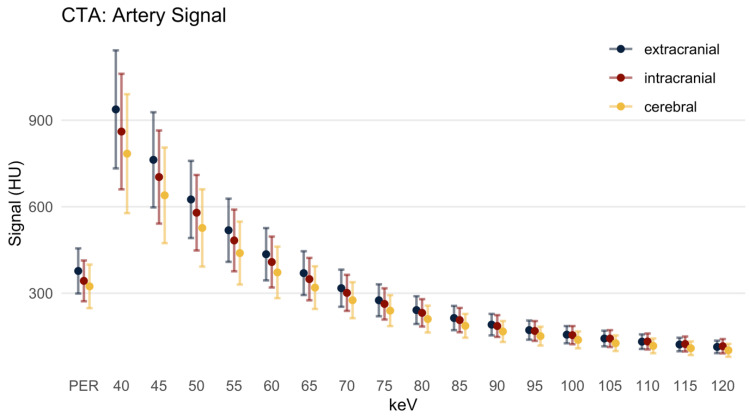
Artery signal in Hounsfield units, mean ± SD. There was a monotonic decrease of the signal with increasing keV level, and the signal of the PER resembled 60–65 keV. HU: Hounsfield units; PER: polyenergetic reconstruction; keV: keV level of the monoenergetic reconstruction. Extracranial: internal carotid artery C1 segment and vertebral artery V3 segment; intracranial: internal carotid artery C4 segment and basilar artery; cerebral: ACA, MCA and PCA.

**Figure 3 diagnostics-12-01306-f003:**
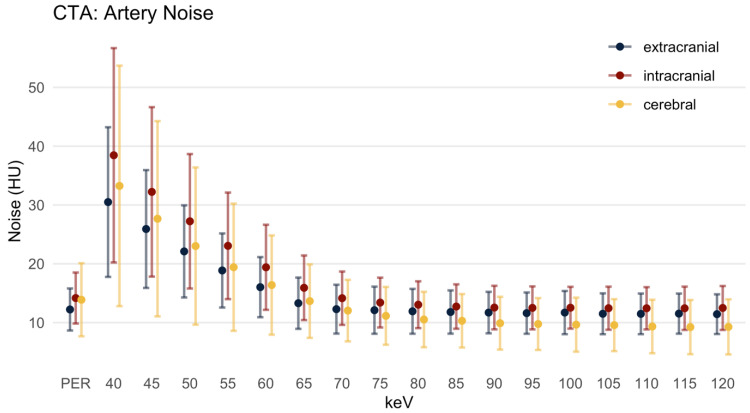
Artery noise in Hounsfield units, mean ± SD. HU: Hounsfield units; PER: polyenergetic reconstruction; keV: keV level of the monoenergetic reconstruction. Extracranial: internal carotid artery C1 segment and vertebral artery V3 segment; intracranial: internal carotid artery C4 segment and basilar artery; cerebral: ACA, MCA and PCA.

**Figure 4 diagnostics-12-01306-f004:**
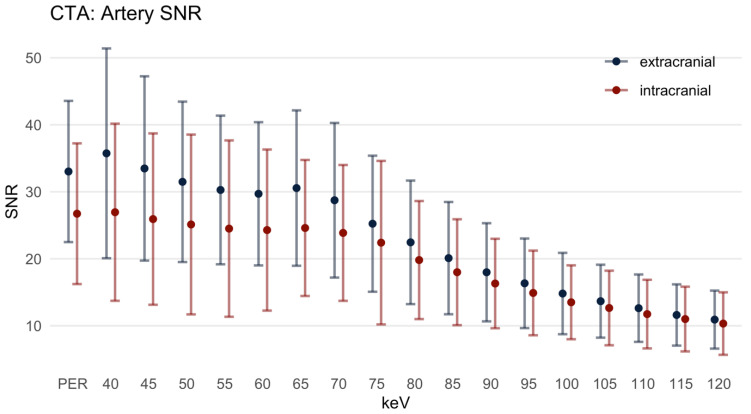
Signal-to-noise ratio, mean ± SD. PER: polyenergetic reconstruction; keV: keV level of the monoenergetic reconstruction. Extracranial: internal carotid artery C1 segment and vertebral artery V3 segment; intracranial: internal carotid artery C4 segment and basilar artery.

**Figure 5 diagnostics-12-01306-f005:**
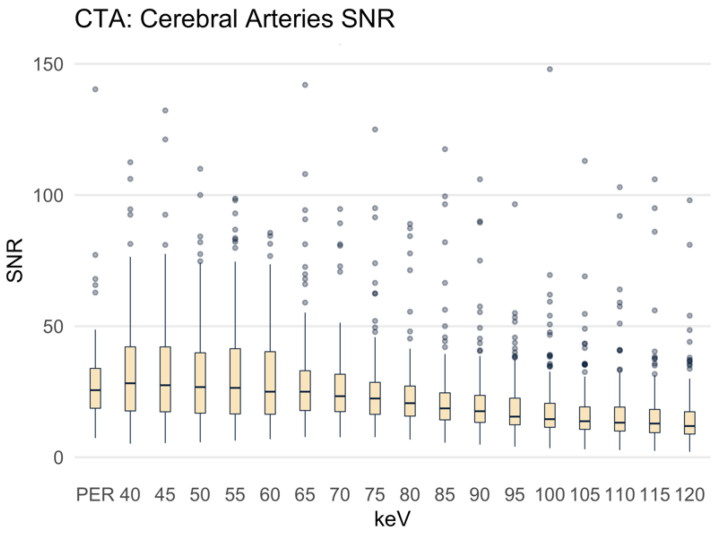
Signal-to-noise ratio presented as boxplots. PER: polyenergetic reconstruction; keV: keV level of the monoenergetic reconstruction. Cerebral arteries: ACA, MCA and PCA.

**Figure 6 diagnostics-12-01306-f006:**
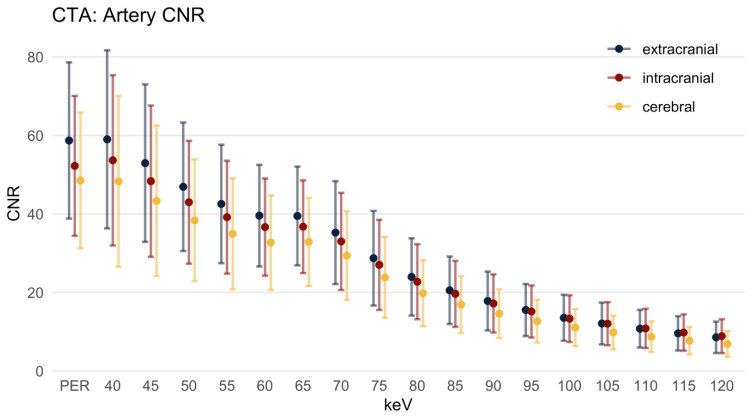
Contrast-to-noise ratio, mean ± SD. HU: Hounsfield units; PER: polyenergetic reconstruction; keV: keV level of the monoenergetic reconstruction. Extracranial: internal carotid artery C1 segment and vertebral artery V3 segment; intracranial: internal carotid artery C4 segment and basilar artery; cerebral: ACA, MCA and PCA.

**Figure 7 diagnostics-12-01306-f007:**
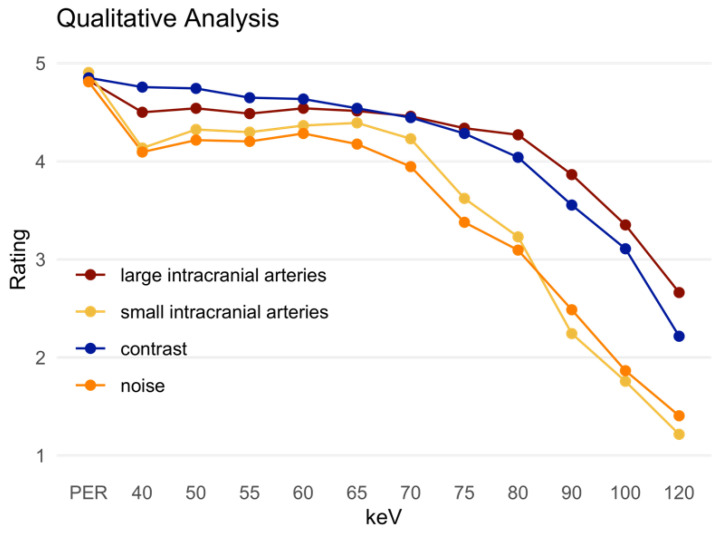
Qualitative analysis presented as the mean of the ratings. PER: polyenergetic reconstruction; keV: keV level of the monoenergetic reconstruction. Large intracranial arteries: assessment of the large intracranial arteries; small intracranial arteries: assessment of the distal cerebral arteries. Contrast: general image contrast. Noise: general image noise.

**Figure 8 diagnostics-12-01306-f008:**
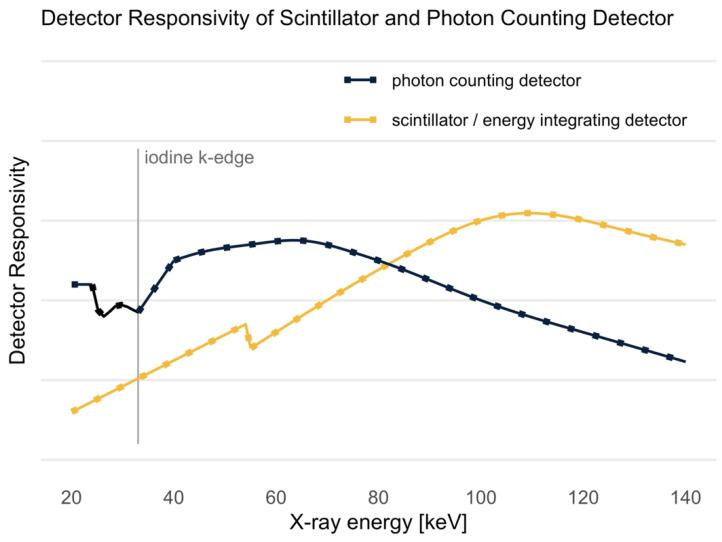
Detector responsivity of scintillator (energy integrating detector) and photon counting detector as a function of the X-ray energy in the X-ray spectrum. The k-edge of iodine is plotted as a vertical line at 33 keV. Figure adapted from Flohr et al. [15].

**Figure 9 diagnostics-12-01306-f009:**
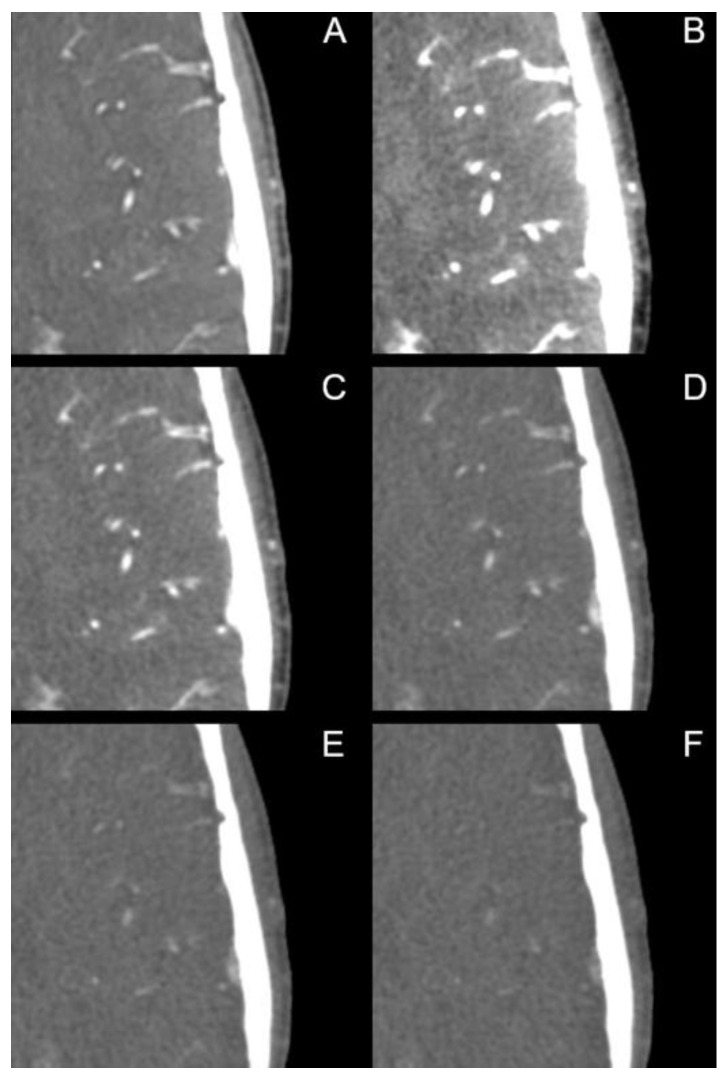
Visualization of small intrasulcal vessels in (**A**) polyenergetic reconstruction and monoenergetic reconstructions with (**B**) 40 keV, (**C**) 60 keV, (**D**) 80 keV, (**E**) 100 keV and (**F**) 120 keV. Window settings were identical in each image. In the qualitative analysis, the polyenergetic reconstruction was particularly superior in the assessment of small vessels.

**Table 1 diagnostics-12-01306-t001:** Comparison of the polyenergetic reconstruction and the best monoenergetic reconstructions.

Extracranial Arteries	PER	MER	*p*-Value
Signal	377.15 ± 78.26 HU	937.55 ± 204.71 HU (40 keV)	*p* < 0.001
Noise	12.23 ± 3.58 HU	11.43 ± 3.38 HU (120 keV)	*p* < 0.001
SNR	33.03 ± 10.54	35.74 ± 15.65 (40 keV)	*p* = 0.003
CNR	58.72 ± 19.90	59.01 ± 22.68 (40 keV)	*p* = 0.447
**Intracranial Arteries**			
Signal	342.76 ± 70.77 HU	860.64 ± 200.45 HU (40 keV)	*p* < 0.001
Noise	14.18 ± 4.34 HU	12.43 ± 3.60 HU (110 keV)	*p* < 0.001
SNR	26.73 ± 10.50	26.95 ± 13.22 (40 keV)	*p* = 0.901
CNR	52.25 ± 17.81	53.67 ± 21.68 (40 keV)	*p* = 0.775
**Cerebral Arteries**			
Signal	323.68 ± 75.67 HU	783.95 ± 206.30 HU (40 keV)	*p* < 0.001
Noise	13.88 ± 6.22 HU	9.23 ± 4.61 HU (115 keV)	*p* < 0.001
SNR	28.60 ± 16.97	40.87 ± 66.94 (50 keV)	*p* < 0.001
CNR	48.53 ± 17.29	48.30 ± 21.75 (40 keV)	*p* = 0.181

PER: polyenergetic reconstruction; MER: monoenergetic reconstruction; SNR: signal-to-noise ratio; CNR: contrast-to-noise ratio.

**Table 2 diagnostics-12-01306-t002:** Results of the qualitative analysis presented as mean ± standard deviation.

Qualitative Analysis	PER	MER	*p*-Value
**Intracranial large arteries**	4.82 ± 0.42	4.54 ± 0.53 (50 keV)	<0.001
**Small intracranial arteries**	4.91 ± 0.41	4.39 ± 0.66 (65 keV)	<0.001
**Contrast**	4.85 ± 0.36	4.76 ± 0.43 (40 keV)	0.057
**Noise**	4.81 ± 0.43	4.28 ± 0.75 (60 keV)	<0.001

PER: polyenergetic reconstruction; MER: monoenergetic reconstruction. Large intracranial arteries: assessment of the large intracranial arteries; small intracranial arteries: assessment of the distal cerebral arteries. Contrast: general image contrast. Noise: general image noise.

## Data Availability

The data are available from the corresponding author on reasonable request.

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
