# Peer review of "Photon Counting CT Angiography of the Head and Neck: Image Quality Assessment of Polyenergetic and Virtual Monoenergetic Reconstructions"

_diagnostics, 2022, doi:10.3390/diagnostics12061306_

Round 1

Reviewer 1 Report

This is a well written manuscript. 

Like it states clearly what statistical tests & software used generating the p-values.

Like it clearly listed step-by-step “Materials and Analysis Methods”

Like it presents both graphs and tables in the results section to help readers easy to understand the results.

The only major concern that I have was the sample size of only using two months of data N=37 (Jan-Feb 2022).  Reference #11 study included 140 patients.  This paper compared to three other studies: 

  1. 2018 published study N=30. Ref #17.
  2. 2014 published study N=41. Ref #18.
  3. 2018 published study N=40. Ref #26.  Suggest if possible, should include more months of data to increase sample size.

Other minor correction:

Page 10 “In comparison to previously published data acquired with DECT scanners, the PCCT 285 appears to offer significant improvements of the image quality of CT angiographies of the 286 head and neck [17,18].” Comment – what test had been performed to support the word of “significant”? If no test performed, suggest only using “better” instead.

Reviewer 2 Report

In this study, the authors introduce and evaluate the novel photon counting CT (PCCT) and its application in CT angiographies (CTA) of the head/neck patients. Based on the two reconstruction methods, PER and MER, the image quality and radiation dose was compared between different parameters of PCCT and conventional dual energy CT (DECT). The authors conclude that the PCCT exhibits better image quality and lower radiation dose. The current study is novel to show the performance of PCCT in clinical application in head/neck imaging. However, some points should be noted and revised to improve the quality of this study.   1. The sample size of this cohort is only 37 patients, which is small to show the detail information of clinical characteristics. Sample size should be increased and it is recommended to add a Table to introduce the demography characteristics of your cohort. 2. This study shows several parameters of image quality, such as CNR, SNR or CTDIvol. However, did the CTA of PCCT matches the clinical diagnosis well? Authors should include and compare the diagnostic data of this cohort, such as specificity or sensitivity of PCCT and DECT, which might exhibit the clinical performance of PCCT well.

Reviewer 3 Report

The authors compared objective and subjective image quality of polychromatic and monochromatic reconstructions of 37 head and neck CTAs obtained with photon counting CT. Their analysis show that overall polychromatic images are of better quality than monochromatic ones.

The major concern with this paper is that because of its design it does not prove the advantage of the PCCT as compared to CT systems with energy integrating detectors. Indeed the authors compare only polychromatic and monochromatic images of PCCT. The comparison to the other study proposed in the discussion is of doubtful utility considering that CNR is affected by many parameters.  In addition, comparing results to only one paper of choice is too limited. This means also that the conclusions drawn by the authors are unsubstantiated based on the results presented.

Furthermore, the authors performed reconstruction with a thickness of 1 mm and an unknown matrix so that one of the main advantages of PCCT, the increased spatial resolution went unexploited.

Changing some reconstruction parameters, as the authors also state, such as the Q level, could change the results. So this might have been something interesting to explore.

Based on the present results, PCCT polychromatic images are better than monochromatic images but it is not proven that these polychromatic images offer any advantage over a conventional CT scanner.

 Another point of concern is the choice to describe together results of arteries adjacent or not to bone. As the authors themselves noted, image quality parameters of these arteries are likely to differ and would deserve a separate analysis.

Other comments:

Methods

Patients with unilateral occlusion or subocclusion were included?

Please indicate all parameters of reconstructions (FOV, matrix)

Statistics:

Please move the classification of intracranial and extracranial arteries in another paragraph. More importantly classification is debatable as arteries next to bones raise different issues as compared to other arteries.

Results:

Page 5 line 177. Remove «initially » or change it to have an appropriate message in english

Page 5 lines 179 to 182. Judging from the graph it is difficult to believe that the signal of the PER is higher than the signal at 65 keV.

Page 5 lines 184 to 186. Were the PER values of intra and extra cranial arteries significantly different or not? This should be indicated

Page 5 lines 188 to 19. See the comment before

Page 6 define the abbreviation EA which has not been defined before

Page 7-8 CNR: please specify what difference are significant and which are not and indicate p values.

Discussion:

Contrast to noise ratio is notoriously affected by many factor, therefore a direct comparison as the one performed by the authors is a stretch. Also the difference the already published paper is not big while the different in standard deviation seems big.

Also if wanting to compare radiation dose other papers should be taken into account where CTA of the head and neck has been performed with dual energy systems with lower radiation dose than in the paper proposed.

All results have to be presented in the results section, including Figures. Please refer to figure 9 in the text.

Page 11 line 360-365 Please cite relevant literature to support your statement such as recent papers proving that PCCT images with voxels of 0.25 are superior to conventional CT in terms of image quality on coronary arteries and cardiac imaging.

https://pubmed.ncbi.nlm.nih.gov/34711766/

https://pubmed.ncbi.nlm.nih.gov/35166583/

Conclusions lines 379-384. This entire part of the conclusion (“The results of our present study 379 underline the great potential of this novel detector technology to improve the image qual- 380 ity of CT angiographies, which may lead to a decisive reduction of radiation exposure and 381 contrast media. The quantitative parameters, as evaluated in the present study, outper- 382 formed data provided for other DECT methods, while at the same time requiring less con- 383 trast medium and far lower radiation dose.”)

 is a big baseless overstatement. The authors have not proven the superiority of PCCT, nor that radiation dose and contrast media can be reduced with this technique.

Figures

Figure 5 . All outliers should be visible or the readers should be provided another way to assess the dispersion of data.

Round 2

Reviewer 2 Report

All the comments were addressed.

Author Response

Dear Reviewer,

thank you again for your time and effort to help improve the manuscript.

Reviewer 3 Report

Dear authors,

Thank you for your revisions. You have addressed most of the points of concern of the previous version.

However, the conclusion of the abstract still needs to be revised. I would suggest to simply remove the last sentence. By removing this sentence, you would be scientifically sound focusing only on things you actually investigated and proved as well as in line with the revised version of the paper.
